



# Past African dust inputs in Western Mediterranean area controlled by the complex interaction between ITCZ, NAO and TSI

Sabatier Pierre[1*], Nicolle Marie[2], Piot Christine[3], Colin Christophe[4], Debret Maxime[2], Swingedouw Didier[5], Perrette Yves[1], Bellingery Marie-Charlotte[1,3], Chazeau Benjamin[1,3,6], Develle Anne-Lise[1], Leblanc Maxime[4], Skonieczny Charlotte[4], Copard Yoann[2], Reyss Jean-Louis[1], Malet Emmanuel[1], Jouffroy-Bapicot Isabelle[7], Kelner Maëlle[8], Poulenard Jérôme[1], Didier Julien[7], Arnaud Fabien[1], Vannière Boris[7]

[1]EDYTEM, Université Savoie Mont-Blanc, CNRS, Le Bourget du Lac, pierre.sabatier@univ-savoie.fr
[2]M2C, Université de Rouen, CNRS, Rouen
[3]LCME, Université Savoie Mont-Blanc, Le Bourget du Lac
[4]GEOPS, CNRS-Université de Paris-Sud, Université Paris-Saclay, Orsay
[5]EPOC, CNRS-Université de Bordeaux, Pessac.
[6]LCE, Université Aix-Marseille, CNRS, Marseille
[7]Chrono-Environnement, Université de Franche-Comté, CNRS, Besançon
[8] SPE, Université de Corse Pascale Paoli, CNRS, Campus Grimaldi, BP 52, F-20250 Corte, France

*Correspondence to*: Pierre Sabatier (pierre.sabatier@univ-savoie.fr)

**Abstract. North Africa is the largest source of mineral dust on Earth, which has multiple impacts on the climate system; however, our understanding of decadal to centennial changes in African dust emissions over the last few millennia is limited. Here, we present a high-resolution multiproxy analysis of sediment core from high-elevation lake Bastani, in Corsica Island to reconstruct past African dust inputs to the Western Mediterranean area over the last 3150 yrs cal BP. Clay Mineralogy and geochemical data allows us to identify that terrigenous fluxes are almost exclusively related to atmospheric dust deposition from the West and North Sahara over this period, which is consistent with current observations. High resolution geochemical contents provide a reliable proxy of Saharan dust inputs with millennial to centennial scale variations. Millennial variations have been correlated to the long term southward migration of the Intertropical Convergence Zone (ITCZ) with an increase of dust input since 1070 yrs cal BP. This correlation suggests a strong link with ITCZ and could reflect the increased availability of dust sources to be mobilized with an increase in wind and a decrease in precipitation over West and North Africa. For centennial to decadal variations, wavelet analyses show that since 1070 yr cal BP, NAO is the main climatic forcing with an increase of Saharan dust input during positive phase, as suggested by previous study over the last decades. However, when ITCZ is in a northern position, before 1070 yr cal BP, wavelet analyses indicate that total solar irradiance (TSI) is the main forcing factor, with an increase of African dust input during low TSI. With climate reanalysis over the instrumental era, during low TSI we observe a significant negative anomaly in pressure over Africa which is known to increase the dust transport. These two climatic**





**forcing factors (NAO, TSI) modulate Saharan dust inputs to the Mediterranean area at centennial timescale through changes in wind and transport pathways.**

## 1 Introduction

The dust cycle is an important part of the Earth system: each year, an estimated 2000 Mt of dust is emitted into the atmosphere (Shao et al., 2011). Mineral dust plays a role in the different components of the climate by affecting the radiative budget and therefore the energy balance of the Earth system. Dust can also act as a cloud condensation nucleus, changing the hydrological cycle by modifying cloud cover and microphysics (Rosenfeld et al., 2008). Mineral dust contributes also to the carbon cycle as an external source of nutrients to the oceans and to remote terrestrial ecosystems (Maher et al., 2010; Pabortsava et al.,
2017; Shao et al., 2011; Yu et al., 2015). North Africa is the world's largest sources of mineral dust to the Earth system and influences spread over both the Atlantic to the American and Caribbean regions and the Mediterranean to Europe (Goudie and Middleton, 2001; Prospero, 2002). Understanding dust emission, transport and deposition requires the study of both past and present climatic variations. Advances in remote sensing and modelling have improved understanding of dust cycle, while improvements in paleo-sciences allow for the reconstruction of both dust record emissions through time and the source region
fingerprinting (Marx et al., 2018). African dust emissions and transports exhibit variability on diurnal to decadal timescales under different atmospheric patterns (Evan et al., 2016). In At the scale of glacial to interglacial climates, dust deposition recorded in marine and continental sediments in mid to high latitudes indicates that dust fluxes have changed greatly over such transitions (Maher et al., 2010). However, except for the African Humid Period (AHP) (deMenocal et al., 2000; Ehrmann et al., 2017; McGee et al., 2013), the influence of the Holocene centennial climate variability on the past dust cycle remains
poorly quantified, while it can be a key element of the forcing of climatic variations and deserve therefore to be integrated in transient climate simulations of the Holocene (Albani et al., 2015). For instance, the neglect of atmospheric dust reduction in early to mid-Holocene in climate models could partly explain the model-data temperature discrepancy in Northern Hemisphere (Liu et al., 2018).

Past dust deposition in the Atlantic (deMenocal et al., 2000; McGee et al., 2013) and Mediterranean (Bout-Roumazeilles et
al., 2007, 2013; Ehrmann et al., 2017; Rodrigo-Gámiz et al., 2011; Wu et al., 2017; Zhao et al., 2016) regions are classically reconstructed from marine sediment cores. The respective influence of fluvial versus eolian input and the low time resolution of marine sedimentary records over the Holocene period do not allow for studies of short-term (decadal to centennial) variability. Other continental archives such as peat bogs (Le Roux et al., 2012; Longman et al., 2017) or lake (Zielhofer et al., 2017) demonstrate the possibility to record African dust. A recent study, in the Iberian Peninsula, shows the potential for high-
elevation lake sediment to record African dust (Jiménez-Espejo et al., 2014), despite the low resolution of the sequence. In this paper, we present a high-resolution African dust record from sediment cores sampled in Lake Bastani, a high-elevation lake (2092 m above sea level) located on Corsica Island in the northwestern Mediterranean Sea (Figure 1), which records the



variability of African dust inputs over the Late Holocene with high temporal resolution allowing for the discussion of centennial climate forcing, filling the gap between long-term instrumental observations and millennial reconstructions.

## 2. Material and methods

### 2.1 Study area

In the Mediterranean area, despite a large daily variability, climatological data show a clear seasonal cycle of African dust transport with a maximum during the dry season: Transport begins over the eastern basin in the spring and spreads over the western basin in the summer (Moulin et al., 1998). This dust mobilization is strongly related to the presence of low pressure systems over North Africa (Moulin et al., 1998). Dust deposition in the western Mediterranean region, including Corsica Island, originates from three main dust emission hot spots: PSA1 (northern Sahara), PSA2 (western Sahara) and PSA3 (Sahel)
(Prospero, 2002; Scheuvens et al., 2013) (Figure 1), as demonstrated by deposition collectors associated with satellite observations and back trajectory analysis (Bergametti et al., 1989; Guieu et al., 2002; Vincent et al., 2016) (Figure 1a).

Lake Bastani (4.2 ha), a high-elevation lake (2092 m above sea level) with a maximum depth of 24 m is located on Corsica Island in the northwestern Mediterranean Sea (Figure 1). The Lake Bastani watershed is restricted (17.3 ha) and mainly
composed of granodiorite and coarse screes, containing no permanent streams and delivering very limited local lithogenic input to the sediments. This lake formed after the last glacial retreat and is delimited northward by glacial deposits. In its southern part a very restricted delta is present and composited by coarse material (coarse sand to gravel).

Three sediment cores named BAS13P1 (58cm), BAS13P3 (47cm) and BAS13P4 (102cm) were sampled in the central part of the lake (42° 03′ 56.65″ N, 9° 08′ 02.79″ E) under 21 m (P4) and 23m (P1 and P3) of water depth (Figure 1b). Samples were
also collected in the watershed in different potential sediment sources (granodiorite, moraine, soil, delta, Figure 1b).

### 2.2 Analytical methods

Core were splited, photographed, and logged in detail, noting all physical sedimentary structures and the vertical succession of facies. The grain size distributions were determined using a Malvern Mastersizer S at a continuous interval of 1 cm for core BAS13P4. Ultrasound was applied to minimize particle flocculation. The core was also sampled at 1-cm steps for core
BAS13P4 to obtain the loss on ignition (LOI) following (Heiri et al., 2001). LOI at 950°C was interpreted to have a carbonate content of below 4% and likely illustrates the uncertainties of this method. The non-carbonate ignition residue (NCIR) was obtained by removing the LOI at 550°C from the initial dry weight. Clay minerals were identified by X-ray diffraction (XRD) every 2 cm for core BAS13P4, using a PANalytical diffractometer at the Laboratoire GEOPS (Université de Paris XI) on oriented mounts of non-calcareous clay sized particles (<2 μm). The oriented mounts were obtained following the methods
described in detail by Colin et al., (1999). Three XRD trials were performed, each proceeded by air-drying, ethylene glycol



solvation for 24 h, and heating at 490°C for 2 h. Semi-quantitative estimates of clay minerals were mainly made according to the position of the (001) series of basal reflections on the three XRD diagrams using MacDiff software. Samples were also analysed by diffuse reflectance Fourier Transform InfraRed Spectrometry (FTIRS) using a Thermo Nicolet 380 spectrometer equipped with a liquid nitrogen-cooled mercury cadmium telluride (MCT) detector and a diffuse reflectance accessory. Each

sample was scanned 32 times at a resolution of 2 cm$^{-1}$ for wavelengths between 4000 and 650 cm$^{-1}$. The complete absence of the peaks systematically associated with carbonates in lake sediments (1300-1560 cm$^{-1}$, 1780-1810 cm$^{-1}$, 2460-2640 cm$^{-1}$) (Rosén et al., 2010) for all the samples indicates that all these sediments are carbonate-free. X-ray fluorescence (XRF) analysis was performed on the surfaces of the split sediment cores at 1-mm intervals, using a non-destructive Avaatech core scanner. The geochemical relative components (intensities), expressed in counts per second, were obtained at various tube settings: 10

kV at 1.5 mA for Al, Si, S, K, Ca, Ti, Mn, and Fe; 30 kV at 1 mA for Cu, Zn, Br, Sr, Rb, Zr, and Pb; 50 kV at 2 mA for Ba; and each run lasting 60 s (Richter et al., 2006). Principal component analysis (PCA) was performed using "R" software. Major and trace elements analyses on sample from the watershed (n=5) and lake sediment (n=10) were conducted by flow injection ICP-MS (traces) and ICP-OES (majors) at the Service d'Analyse des Roches et des Minéraux (SARM, Nancy, France) using the method described by (Carignan et al., 2001). Each element has a different range of uncertainties, with an average from 1–

5% for major elements and 5–10% for trace elements.

## 2.3 Dating

A continuous sampling step of 5 mm was applied, over the first 10 cm of BAS13P3, to determine $^{210}$Pb, $^{226}$Ra, and $^{137}$Cs activities using well-type, germanium detectors placed at the Laboratoire Souterrain de Modane following Reyss et al., (1995). In each sample, the $^{210}$Pb excess activities were calculated by subtracting the $^{226}$Ra-supported activity from the total $^{210}$Pb

activity. Age model are computed with *serac* R package (https://github.com/rosalieb/serac, Bruel and Sabatier, personal communication). Five $^{14}$C measurements of the terrestrial organic macroremains sampled in core BAS13P1 (4 ages) and BAS13P4 (1 age) were carried out by an Accelerator mass spectrometer (AMS) at the Poznan Radiocarbon Laboratory. The calibration curve IntCal13 (Reimer et al., 2013) was used for the $^{14}$C age calibration (Table 1). Then, we used a smooth spline interpolation with the R-code package clam of R software to generate the age model (Blaauw, 2010).

## 25   2.4 Statistical analysis

Different strategies of multivariate data analysis are used to interpret geochemical data from both sediment core and watershed samples. Principal component analysis (PCA) is performed on major elements and loss of ignition (LOI) from the BAS13 core and sample from the watershed after "centered logratio transformation" (clr-transformation, Aitchison et al., 2002)). These statistical calculations were conducted with "R" software using the package "compositions" and allow us to avoid the so-called

closure operation on the covariance matrix for major element ((van den Boogaart and Tolosana-Delgado, 2008)). Classic PCA was conducted on XRF analysis.





The R-package "segmented" (Muggeo, 2008) is used for break-point analyses on a given linear regression model, segmented estimate a new model having broken-line relationships with the variables specified. A segmented (or broken-line) relationship is defined by the slope parameters and the break-point where the linear relation changes.

Wavelet analysis (WA) is used to decompose a signal into a sum of small wave functions of a finite length that are highly
localized in time for different exploratory scales (Torrence and Compo, 1998). WA corresponds to a bandpass filter that decomposes the signal on the basis of scaled and translated versions of a reference wave function. The "Morlet" wavelet was chosen as the wavelet reference. Several types of wavelets are available, but the Morlet wavelet offers a good frequency resolution and is used most of the time with a wavenumber of 6, for which the wavelet scale and Fourier period are approximately equal. All the series were resampled at the initial point number $n$ of the series.

Because WA is sensitive to large, long-term fluctuations that can occur in paleoclimate series, which may mask fluctuations expressed at the highest frequencies, all series were detrended before performing WA. The long-term trends were calculated with autosignal software using cubic model. The cubic baseline is fitted in a single step matrix solution using a least-squares minimization. The resulting plot of the WA, called the local wavelet spectrum, allows for the description and visualization of the power distribution (z-axis), according to the frequency (y-axis) and time (x-axis). All series were zero-padded to twice the
data length to prevent spectral leakages produced by the finite length of the time series. Zero-padding produces edge effects: the lowest frequencies and near the edges of the series are underestimated, and fluctuations that occur in this area must be interpreted with caution. This area is known as the cone of influence. Monte-Carlo simulations were used to assess the statistical significance of the detected fluctuations. All the detected fluctuations are statistically tested at the $\alpha = 0.05$ significance level against an appropriate background spectrum. Autoregressive modelling was used to determine the AR(1)
stochastic process against which the initial time series was to be tested. For this study, AR(1) corresponds to a red noise of (AR(1)>0).

Cross wavelet (XWT) and wavelet coherence (WTC) were used to examine the relationship between Dust proxy in the sediment and external forcing. Following Grinsted et al., (2004) this analyse allow to test whether forcing can change over the time for specific frequencies. XWT and WTC were calculated using Matlab (R2018) wavelet toolbox. To avoid edge effects,
time series were downsampled at the closer dyadic size of the shortest time serie. Such as for WA, Morlet wavelet (m=6) was used because of its compromise between accuracies both in time and frequencies. XWT gives a cross correlation between both signals, while WTC informs about the phase between them. Right and left arrows are respectively in phase or in phase opposition, up and down arrows show respectively the first or the second signal which predates the other one. Dyadic resampling of the time series denoises the signal allowing to use a first order derivative for detrending. Phase arrows were
shown for correlation superior to 0.7.





### 2.5 Observation-based data pressure data

To evaluate the link between solar forcing and changes in wind patterns over the recent period, we used the twentieth century reanalysis NOAA (20CR) Project version 2 (Compo et al., 2011), consisting of an ensemble of 56 realizations with 2° x 2° gridded 6-hourly weather data from 1871 to 2010. Each ensemble member was performed using the NCEP/GFS atmospheric
model, prescribing the monthly sea surface temperature and sea ice changes from HadISST as boundary conditions, and assimilating sea level pressure data from the International Surface Pressure Databank version 2.0 (http://rda.ucar.edu/datasets/ds132.0). We used the ensemble mean to perform our analysis. We have computed a linear regression between sea-level pressure variations in each grid point and variation of the TSI (Lean, 2009) over the period 1870-2009. We have applied Student's t-test to only show the correlation significant at the 95% level.

## 3 Results

### 3.1 Sedimentology and mineralogy

The sediment consists of olive-grey silty clay with a high organic content (15.7%), up to 20% over the last few centimetres (Figure S1). The carbonate content estimated by LOI950 presents a very low value (<4%, close to the method uncertainties) and infrared analyses show that there is no carbonate present in this sediment (data not shown). The NCIR is relatively constant
and up to 80%, with short variation over the upper 10 cm. This NCIR mainly composed of biogenic silica (diatom frustules observed in smear slices) and terrigenous input from both the watershed and or the eolian flux. The grain-size distribution presents a homogenous content (median (D50) = 32 ± 8 μm) (Figure S1); thus further analyses could not be influenced by grain size variation. On average, the clay mineral fraction is composed of illite (28 ± 5 %), smectite (25 ± 7 %), kaolinite (24 ± 4 %), chlorite (12 ± 4 %) and palygorskite (12 ± 4%) and presents large variations (Figure S1). Toward the top of the core
Illite, kaolinite and chlorite present an increasing trend, with a stabilization for kaolinite, while, smectite and palygorskyte a decreasing one, these tendencies are less marked over the upper 50 cm of the core. All these clay mineral contents present short term variations at pluri-centimeter scale (Figure S1).

### 3.2 Geochemistry

The three cores were correlated using XRF data (Figure S2) to provide a composite sequence of 102 cm. Major and trace
elements were measured using an XRF core scanner and were subjected to PCA (Figure S3); to constrain sediment end-members (Sabatier et al., 2010). Dim1, Dim2 and Dim3 explain 50.5%, 14.9% and 12.1% of the total observed variance, respectively (Figure S3). This PCA allows for the identification of three geochemical end-members: (i) Al, K, Fe, Ti, Ca, Rb, Sr, Ba, Zr, and (Si), which are related to terrigenous inputs (eolian and/or watershed); (ii) S and Br, which are linked to the organic matter content (Bajard et al., 2016); and (iii) a Pb source that may be correlated with periods of past metallurgic




activities (Elbaz-Poulichet et al., 2011). We can note that the Si content appeared to be slightly different from other terrigenous elements because Si is high in aluminosilicate and in diatom content linked to the planktonic productivity in the lake.

Major element concentrations obtain from quantitative analysis on sediment core and samples from the watershed where subjected to PCA after "centered logratio transformation" (van den Boogaart and Tolosana-Delgado, 2008) (Figure 2a). Dim1, Dim2 explain 81% and 13% of the total observed variance respectively and allows for the identification of two geochemical end-members: (i) $Fe_2O_3$, $TiO_2$, $MgO$, $Al_2O_3$ correlated to lake sediment samples and (ii) $CaO$, $K_2O$, $Na_2O$, $MnO$ correlated to sample from the watershed; $SiO_2$ appeared in intermediate position (Figure 2a), as for PCA on XRF data (Figure S3). With this two main end-members identification, Fe/Ca vs Ti/Ca and Fe/K vs Ti/K element ratio from quantitative measurement are then used to understand the geochemical signature of lake sediment (Figure 2b). High resolution XRF Fe content presents i) a main tendency with an increase between ~40 cm to 10 cm and then a decrease and ii) short term increase in Fe contents such as ~8, 13, 18, 25, 33, 38, 45, 52, 59, 68, 73, 85 and 95 cm (Figure 2c). $Fe_2O_3$ concentrations match very well with Fe contents obtained by XRF core scanner with the same variations (Figure 2c) and the correlation between these two type of data (r=0.86, p < 0.0013) allows us to use Fe content from XRF data as a high resolution quantitative measurement.

### 3.3 Chronology

The excess [210]Pb downcore profile showed a regular decrease punctuated a phase of by the homogenized [210]Pbex activities (Figure S4, grey band), which corresponds to the mass wasted deposit with a constant Fe content (cps) identified in (Figure S2) and to the higher NCIR content (Figure S1). Following Arnaud et al., (2002), event was excluded from the construction of an event-free sedimentary record because they were considered as instantaneous deposits. [210]Pbex activities plotted on a logarithmic scale revealed two linear trends providing two mean sedimentation rates of $0.79 \pm 0.1$ mm yr[-1] above 3.5 cm and $0.23 \pm 0.02$ mm yr[-1] below 6.5cm Figure S4). Ages were then calculated using the CFCS model applied to the original sediment sequence to provide a continuous age-depth relationship. The [137]Cs profile presented one peak at 6.75 cm which corresponds to maximum nuclear weapon tests in the Northern Hemisphere in 1963 CE. The Chernobyl accident in 1986 CE is not clearly visible on this profile. This age model is confirmed by the correlation between the [210]Pbex and the [137]Cs peak (Figure S4).

Sediment core chronology is based on short-lived radionuclide data (Figure S4) and 5 radiocarbon dates (Table 1), both correlated to the BAS13P4 core (Figure S2). Age modelling provided a precise age model over the last 3150 yrs cal BP, with an average sedimentation rate of 0.34 mm.yr[-1] (Figure 3), using clam software (Blaauw, 2010). This age model is confirmed by three Pb peaks at the Roman Period (with a double peak centred at 270 and 120 yrs cal BC); the Medieval Period (1120 yrs cal AD); the modern industrial period starting at 1870 yrs cal AD with a maximum between 1970 and 1980 yrs cal AD (Figure 3) related to the Pb additives in gasoline. The two older Pb peaks correspond to well-known past metallurgic activities in the Mediterranean area (Elbaz-Poulichet et al., 2011) and demonstrate that the age model is well defined and that this lake system is ideal for recording atmospheric inputs.



## 4 Discussion

### 4.1 Sediment sources

In northern Africa, the ratios of clay minerals, such as illite to kaolinite (I/K ratio) and chlorite to kaolinite (C/K ratio), can serve as "fingerprints" of specific source areas (Bout-Roumazeilles et al., 2007; Caquineau, 2002; Guieu et al., 2002) (Figure 1). The fibrous clay mineral (palygorskite), typical of arid to semiarid areas, appears to be a suitable source marker for the north and western Sahara (PSA1 and PSA2). The 3150 yrs cal BP Lake Bastani sediment record reveals a relatively high percent of palygorskite in clay mineral fraction; therefore, African dust deposits constitute a major part of the fine terrigenous flux in this lake (Figure 4a). The I/K ratio is typical of the Saharan sources but not the Sahelian source (Figure 4b). The low value of the C/K ratio is typical of the western Saharan source (PSA2), the most recent samples indicate intermediate sources between western and northern Saharan sources (Figure 4c). The decreasing trend in palygorskyte content probably reflect progressive trend with more input from PSA1 over, as also suggested by the increasing trend of C/K. This result is consistent with the modern dust provenance area identified for Corsica Island through air mass trajectories, with the dominant dust flux originating from the western (>30%) and northern (25-30%) Sahara (Vincent et al., 2016).

The NCIR (eolian, watershed, biogenic silica) flux for the last century in Lake Bastani is 4.5 g.m$^{-2}$.yr$^{-1}$. In comparison, the longest deposition dust fluxes measured in Corsica Island (10 years) vary between 4 to 26 g.m$^{-2}$.yr$^{-1}$ (M.D. Loÿe-Pilot and Martin, 1996), with the lower value of this range corresponding to the NCIR flux calculated from lake Bastani sediment core. Even a low dust deposition year (2013 : 2.1 g.m$^{-2}$.yr$^{-1}$) (Vincent et al., 2016) corresponds to half of the NCIR flux. Thus, even though eolian input may vary from one year to another, dust fluxes measured in Corscia Island may represent almost all of the silicate source to this system and siliceous input from the watershed can be neglected in the variability of our reconstructed signal.

Geochemical data allow the identification of two mains terrigenous end-members with lake sediment characterised by Fe$_2$O$_3$, TiO$_2$, MgO, Al$_2$O$_3$ while the watershed is mainly composed by CaO, K$_2$O, Na$_2$O, MnO. We used Fe/Ca vs Ti/Ca and Fe/K vs Ti/K on samples from of the lake sediment and watershed associated to geochemical data of dust deposits in NW Mediterranean (Spain, Avila et al., 1997, 2007) and Central Mediterranean areas (Adriatic Sea, Italia, Sicily Tomadin et al., 1984) (Figure 2b). These data allow to identify that geochemical composition of lake sediment is similar to non-carbonated dust samples with no evident contribution of watershed end-member to the lake sediment, in agreement with previous finding on fluxes. From XRF measurement on lake sediment, terrigenous elements present in the African dust, such as Ti and Fe are correlated (r=0.72, p < 10$^{-16}$), but the Fe signal is less noisy. Moreover, Fe and Ca, K contents present the same variations in lake sediment indicating that even for terrigenous elements typical from the watershed (Ca, K Figure 2b), most of their variations is influenced by African dust, in good agreement with NCIR flux estimation. This very low contribution of terrigenous elements from the watershed precluded the Fe/Ca or Fe/K ratio as proxy of dust input. In African dust, iron is present in Fe-containing silicate, iron oxide and hydroxide coating and cement on grains and in the lattice of clay minerals (Formenti et al., 2011; Scheuvens et al., 2013). Considering that 1) the high Fe content in African dust, 2) the negligible input from the watershed, 3) no grain-size





variations in the Lake Bastani sediment (SI1), 4) low sedimentation rate changes (Figure 2) and 5) good correlation between XRF core scanner and $Fe_2O_3$ concentrations, the Fe content measured by XRF has been used to reconstruct African dust input variations through time (Figure 4d). Ca is classically used as tracer for carbonate minerals (calcite and dolomite) and is present in African dust; however, carbonate was not identified in the Lake Bastani sediment using infrared spectrometry (data not

shown) and LOI. The absence of carbonate in the sediment could be linked to calcite dissolution during both atmospheric transport (Avila et al., 2007) and deposition in the lake water.

## 4.2 African dust input and climate forcing

### 4.2.1. Millennial variation

A high-resolution (3 yrs) Fe content over the last 3150 yrs cal BP show a long-term variation with a significant increase from

approximately 1000 yrs cal BP and centennial scale variations, as well as an increase during the Little Ice Age (650-50 yrs cal BP) (Figure 4d). This increase in dust input is earlier than previously published by Mulitza et al., (2010) in Atlantic area which the authors attributed to human contribution through changes in the agriculture in Sahel region (beginning of the nineteenth century). Reconstructed African dust input present also high short term variation at centennial to decadal timescale. Over the last century, we observed an African dust increase that reached a maximum between 1945-1975 and then a decrease until

today, like the decrease trend observed by Evan and Mukhopadhyay, (2010). African dust input variations on Corsica are potentially linked to the availability of dust sources that could be mobilized and to the transport pathway that allows dust to reach this island.

We calculate the long-term trend (centennial to millennial) using the cubic model which allows for reconstruction of long-term dust signal in Lake Bastani. This millennial scale variation shows an increasing trend, similar to the long-term southward

migration of the ITCZ (Haug et al., 2001) (Figure 4d, e) in response to a gradual orbitally-induced decrease in Northern Hemisphere insolation. These two long-term reconstructed trends present a good correlation ($R^2$=0.90, p-value< 2.2e-16, F-statistic = 4857). A break-point analysis on linear regression model between the two initial signal shows a change at 1070 yrs cal BP (Figure 4) with a positive trend since that time. Modern increases in boreal winter and summer dust emissions are correlated with a southerly position of the ITCZ over North Africa, which are associated with increased surface winds over

central and western North Africa, (Doherty et al., 2012, 2014). The annual dust cycle may be partly explained by seasonal changes in the positions of the Intertropical Convergence Zone (ITCZ) and associated rainfall distributions. The most Northern position of the ITCZ is responsible for the rainy (monsoon) season, rainfall reduces the atmospheric dust content by both increases soil moisture and vegetation cover reducing dust emission from the ground and cleans the atmosphere by removing dust particles (wet deposition) (Engelstaedter et al., 2006). This correlation could reflect the availability of dust sources for

mobilization, suggesting a long-term forcing of the southward ITCZ migration on the increase dust export to the Mediterranean area since 1070 yrs cal BP. Over a longer period, such link is also observed with a drastic decline in dust flux occurred during the AHP (11,7-5 kyrs) related to a combination of decreased wind and increased precipitation, which is associated with a



northward shift of the ITCZ (Bout-Roumazeilles et al., 2007; McGee et al., 2013; Rodrigo-Gámiz et al., 2011) in relation to the decrease in summer orbital insolation (deMenocal et al., 2000).

### 4.2.2. Centennial variation

A recent study, based on a comparison between dust time series and projections of the wind pattern onto climate models shows that wind intensity is an important factor controlling African dust variability over the last several decades, and this explains why time series of dust are correlated with diverse climate phenomena such as the NAO, ITCZ, and Sahelian drought (Evan et al., 2016). To test such relationships over the last millenniums, wavelet analyses have been performed on the Bastani dust inputs, total solar irradiance (TSI) (Steinhilber et al., 2012), ITCZ (Haug et al., 2001), ENSO (Moy et al., 2002) and North

Atlantic Oscillation (NAO) (Franke et al., 2017) reconstructions. They highlight significant cyclicities (up to more than 95% level of confidence, Figure 5). The scalogram of dust inputs (Fe) presents three different cycles of approximately 200, 300 and 450 years (Figure 5a). The power spectrum of the ITCZ and ENSO reconstructions do not show significant cycles on this period, except between 2500-3100 yrs cal BP for the ITCZ reconstruction with a period of approximately 200 years common with the dust signal (Figure 5b) with large oscillation in the ITCZ position (not shown for ENSO). We use a recent NAO

reconstruction for the last 2000 years (Franke et al., 2017), based on 37 high resolved proxy records following the recommendation of Ortega et al., (2015), which showed that an NAO index reconstruction based on two proxies is not sufficient to define long-term NAO variability. While this record is only covering the last 2 millennia, wavelet analyses have been performed and display cycles of approximately 200 and 300 years (Figure 5d). Cross wavelet analysis between dust input and NAO signal over the last 2000 years' highlights a period of high (low) dust inputs in Corsica corresponding with positive

(negative) NAO index over the last millennium (Figure 6a). During positive NAO phase, pressure gradient between the Icelandic low and the subtropical high is more intense than normal and westerly winds are stronger across northern Europe, which is associated with mild temperatures and higher precipitation, while dryer conditions than usual are produced at lower latitudes across southern Europe. Moulin et al., (1997) showed that, over the modern period, the positive NAO phase, with drier conditions over southern Europe, the Mediterranean Sea and northern Africa, induced higher dust transport and affected

both the pattern and intensity of the transport of African dust. When the NAO is negative, the pressure gradient decreases and the westerlies are shifted to the South providing precipitations over the Mediterranean and the North African continent, restricting dust uptake and transport (Moulin et al., 1997). The correlation between positive summer NAO and high dust contribution has been identified more recently in a 11 years (2001-2011) record (Pey et al., 2013). But, even if in our record we compare dust signal (mainly spring and summer) with yearly index of the NAO, this strong relation appeared since ~1000

yrs cal BP (Figure 6a), which correspond to the period of long-term increase in African dust, correlated with the southward ITCZ migration (Figure 4), suggesting that long-term forcing through ITCZ migration have an impact on the NAO/African dust correlation. This influence can be explained by the fact that the ITCZ and westerlies are both linked to the Hadley Cell through North Atlantic subtropical high, and the position of the westerlies are influenced by the NAO (Souza and Cavalcanti,



2009). During the positive phase of the NAO with strong North Atlantic subtropical high, the ITCZ is displaced southward in April (Souza and Cavalcanti, 2009). Here we suggest that a long-term southward position of the ITCZ over the last millennium may have enhanced NAO forcing on African dust inputs to the Western Mediterranean area in spring and summer.

The scalogram of TSI reconstructions shows significant periodicities already known from cosmic-ray-produced radionuclide records (Steinhilber et al., 2012), such as the period of ~200 yrs corresponding to the de Vries cycle as well as the period ~300 yrs and a lesser extend a less significant one around 430 years. These two significant cycles are the same as the two shorter cycles observed in the dust inputs (Figure 5a, b). Cross wavelet analysis between dust input and TSI signal over the last 3000 years' highlights a period of high (low) dust inputs in Corsica corresponding with low (high) TSI between 3000 to 1000 yr cal
BP with a shift from the 200 to the 300 yrs cycles around 2000 yr cal BP (Figure 6b). Variation in the solar UV radiation affects the stratospheric ozone, leading to temperature variations in the top stratosphere. The resulting temperature gradients lead to changes in the zonal wind, which change the planetary wave–mean flow interactions and appear to be an indirect effect of the changes in the global atmospheric circulation through the `top-down' mechanism (Gray et al., 2010). Thus, large changes in solar radiation indirectly affect the climate by inducing atmospheric changes, such as in the large European Atlantic sector
with wind and precipitation changes (Martin-Puertas et al., 2012), or in the Mediterranean area, which results in both increased storminess (Sabatier et al., 2012) and a higher flood frequency (Sabatier et al., 2017; Vannière et al., 2013).

To explain the link we found between TSI and dust records, we argue that changes in the TSI can (slightly) modify the main wind patterns. Indeed, a decrease in TSI diminishes the surface solar insolation in the tropical area, especially in cloud free areas, which can decrease evaporation there, and then the moisture transport into the precipitation convergence zone, leading
to a weakening of the Hadley Cell, as illustrated in some model simulations (Meehl et al., 2004). Such a weakening can then induce in the subsidence zone a decrease in the magnitude of the anticyclones over North Africa, i.e. a negative sea-level pressure or low-pressure anomaly. Such a low-pressure anomaly is known to increase the dust transport over the recent period (Moulin et al., 1998). To support this link between the TSI and changes in pressure pattern over North Africa, we performed a regression analysis over the instrumental period, using TSI reconstruction (Lean, 2009) and the NOAA 20CR reanalysis
(Compo et al., 2011). We find in spring a significant negative anomaly in pressure when TSI is decreasing (Figure 7), which is going in line with the mechanism depicted before. Moreover, this mechanism could be also associated to Saharan Heat Low (SHL), an area of high surface temperature and low surface pressure within the summertime Sahara Desert, when SHL cools, wind speed increases over the Sahara and increasing dust emission from the major dust sources in the Sahara is observed (Wang et al., 2015, 2017). Therefore, we argue that such a relatively simple process, observed over the last century, already
found in model simulations concerning the weakening of the Hadley Cell (Meehl et al., 2004) can explain the link we found between low TSI and higher dust inputs records in Corsica for the period between 3000 to 1000 yrs cal BP. Thus, we hypothesize here that when ITCZ is more in a northern position (before 1070 yrs cal BP) such mechanism is enhancing through changes in wind intensity and transport pathways of dust into the Western Mediterranean area.



## 5 Conclusions

High-elevation sediment record of lake Bastani on Corsica Island in the northwestern Mediterranean Sea allows for a reconstruction of high-resolution African dust variability over the last 3150 yr cal BP. Thanks to mineralogical and geochemical analyses we can unambiguously identify that African dust is the main terrigenous sediment component. This
multiproxy approach reveals that both western and northern Sahara are the main African dust sources to the Western Mediterranean area in agreement with the modern dust provenance area identified for Corsica Island through air mass trajectories. We suggest that the millennial scale variations of Saharan dust inputs have been forced by the southward migration of ITCZ with an increase since 1070 yrs cal BP in response to a gradual orbitally-induced decrease in Northern Hemisphere insolation. This correlation could reflect the increased availability of dust sources to be mobilized by an increase in wind and
a decrease in precipitation, soil moisture and vegetation cover over north Africa. At centennial timescale, NAO and TSI are the two main climate forcing identified but their respective influences seem to be related to the ITCZ migration. Since 1070 yr cal BP (ITCZ southern position) the NAO is the dominant climate forcing with an increases of Saharan dust input during NAO positive phase with drier conditions over southern Europe, the Mediterranean Sea and northern Africa affecting both the pattern and intensity of the transport of African dust, as observed from the instrumental period. Between 3150 and 1070 yr cal
BP (ITCZ in more northern position) the centennial increases of Saharan dust inputs are correlated to low TSI. During the instrumental era, we observe during low TSI a significant negative anomaly in pressure over Africa which is known to increase the dust transport over the recent period. We suggest that when ITCZ is more in a northern position such atmospheric mechanism is enhancing and induces changes in wind intensity and transport pathways of dust into the Western Mediterranean area. As long-term ITCZ migration reflects the availability of dust sources to be mobilized; NAO and TSI modulate short term
Saharan dust inputs to the Western Mediterranean area through changes in wind and transport pathways. The highlighting of these three climatic controls on dust inputs at different timescales allows for a better definition of the dust-climate interactions from a long-term perspective.

**Acknowledgments:**

We thank the Office Environemental Corse (OEC, Gwenaëlle Baldovini and Pierre-Jean Albertini), the Parc Naturel Regional
de Corse and the Direction regionale de l'Environnement, de l'Aménagement et du Logement de Corse (DREAL) for the coring authorization and support during fieldwork. The authors would like to thank Marie-Dominique Loÿe-Pilot for its manuscript reading, Victor Froissard for help with R-code and Jérôme Debret for his help during the field trip. Part of the [14]C analyses were acquired thanks to the CNRS-INSU ARTEMIS national radiocarbon AMS measurement programme at Laboratoire de Mesure 14C (LMC14) in the CEA Institute at Saclay (French Atomic Energy Commission). The figure about
short-lived radionuclides was computed thanks to a code developed by Rosalie Bruel (manuscript in prep). The authors would also like to thank the Laboratoire Souterrain de Modane facilities for the gamma spectrometry measurements and Environnement, Dynamique et Territoires de Montagne for the X-ray fluorescence analyses. **Funding:** this research is part of



the HoTMED project (lead by Boris Vannière) and funded by the Région Franche-Comté through the University of Franche-Comté and by the CNRS through the PaleoMEx-INEE program.

**Author Contribution:** P.S., B.V., M.D. and C.P. designed research; P.S., C.C., E.M., M.C.B., B.C., J.-L.R., A.-L.D., Y.C., J.D., I.J.B., C.P., D.S., M.K. and J.P. performed research; P.S., M.N., Y.P., C.C., B.V, D.S., C.S., M.L. and M.D. contributed new reagents/analytic tools; P.S., Y.P., M.N., B.V., D.S., C.S., M.L. and M.D., analysed data; and P.S., M.N., C.C., M.D., B.V., D.S. and F.A. wrote the paper.

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





| Samples | Material dated | Cores | Composite depth (mm) | Uncalibrated age (BP) | Uncertainty | Calibrated age ranges at 95% confidence interval (cal. BP) |
|---|---|---|---|---|---|---|
| Poz-69623 | O. Macro | BAS13P1 | 118 | 325 | 30 | 307-468 |
| Poz-69624 | O. Macro | BAS13P1 | 226 | 545 | 30 | 516-635 |
| Poz-73333 | O. Macro | BAS13P1 | 380 | 1410 | 30 | 1286-1358 |
| Poz-69625 | O. Macro | BAS13P1 | 566 | 2110 | 30 | 1996-2152 |
| Poz-61153 | O. Macro | BAS13P4 | 1020 | 3000 | 35 | 3076-3327 |

**Table 1:** List and depth reported in the composite depth of 14C dates for the Lake Bastani cores. The composite depth is obtained from Figure S2.



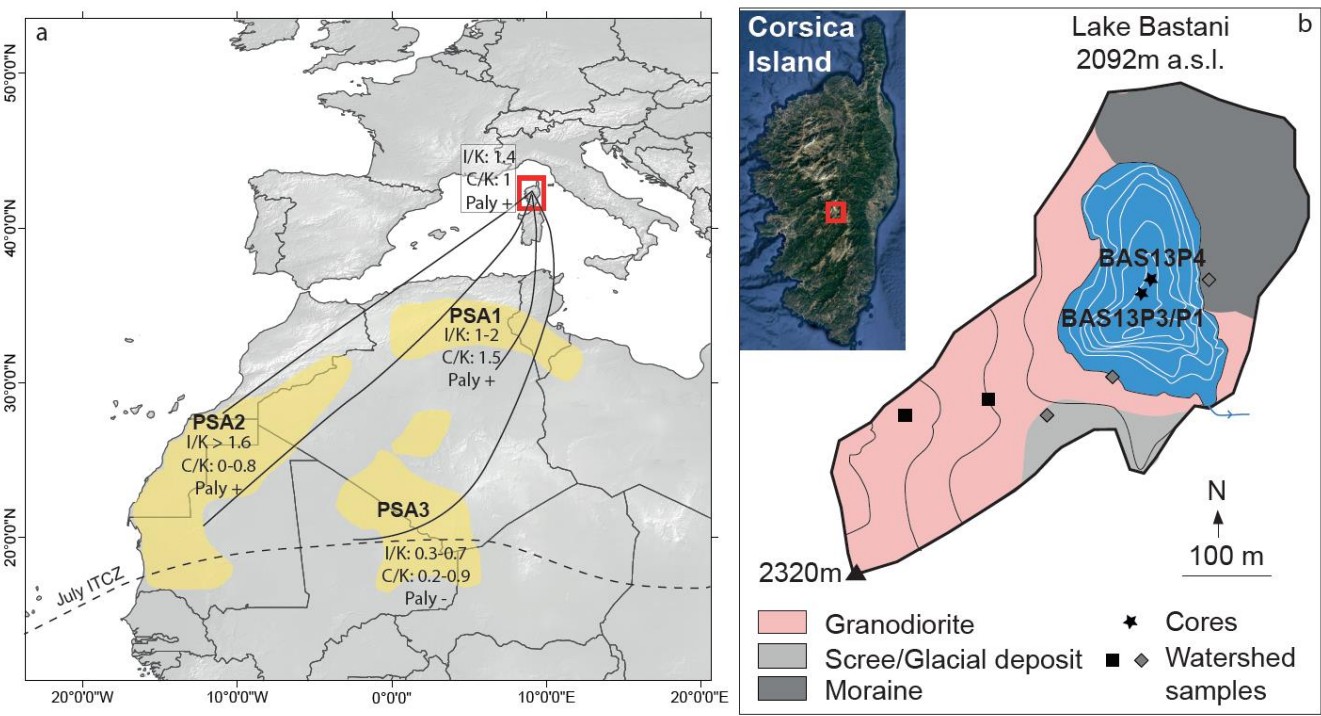

**Figure 1.** (a) Map of the western Mediterranean area with major potential source area for northwestern African dust (PSA1, PSA2, PSA3), with a clay mineral ratio (illite to kaolinite and chlorite to kaolinite) and palygorskite content (Formenti et al., 2011; Scheuvens et al., 2013). Black curves are the main back trajectories for the dust event collected on Corsica Island (study area, red rectangle) modified from Guieu et al, (2002). The dotted line represents the summer (July) ITCZ position. (b) Aerial photography of Corsica Island with the localization of Lake Bastani. The Lake Bastani bathymetry map and its watershed with the localization of the three studied cores (BAS13P1, BAS13P3, BAS13P4) and samples on the watershed.





**Figure 2**: (a) PCA biplot on major elements obtained by ICP-MS measurement corrected for LOI content and after centered logratio transformation. Data use in this PCA are from both lake sediment (dotted area) and watershed samples (granodiorite and quaternary deposits). (b) selected element ratios (Fe/Ca vs Ti/Ca and Fe/K vs Ti/K) from ICP-MS measurement on lake sediment, watershed samples and data form dust deposits in NW Mediterranean (Spain, Avila et al., 1997, 2007) and Central Mediterranean areas (Adriatic Sea, Italia, Sicily Tomadin et al., 1984). (c) Comparison between Fe content form XRF core scanner and ICP-MS analyses.



**Figure 3.** Age-depth model of the 102-cm Lake Bastani sedimentary sequence (BAS13P4), including $^{210}$Pb/$^{137}$Cs chronology (Figure S4) and $^{14}$C ages (Table 1). Lead measurement in the lake sediments with the identification of three pollution peaks and associated ages.







**Figure 4.** (a) Palygorskite content (%). (b) I/K illite to kaolinite ratio to discriminate source area of Sahel (PSA3) from the Sahara (PSA1 and PSA2). (c) C/K chlorite to kaolinite ratio to discriminate source areas in North Sahara (PSA1) from West Sahara (PSA2). Vertical dashed lines represent the upper and lower boundary of African sources. (d) Fe content interpreted as total dust input. (e) variation of the ITCZ position (Haug et al., 2001), reverse scale. Red curves present the long-term trend, and black arrows indicate a common variation. The horizontal gray line at 1071 cal BP represents this break-point analysis on the linear regression model between Fe content (dust input) and ITCZ.









**Figure 5**: Individual wavelet analyses for a) dust content (Fe, Lake Bastani), b) TSI (Steinhilber et al., 2012), c) ITCZ (Haug et al., 2001) and d) NAO+ (Franke et al., 2017) signals. NAO signal cover the last 2000 yrs. Occurrence of the periods (labelled in white) with respect to the time is given by the bright yellow-red colors. The 95% confidence levels are indicated with the dotted line.





**Figure 6**: Cross wavelet analyses for dust input versus NAO (a) and TSI (b). Dotted line represented identified significant cycle for each signal in single wavelet analyses (Figure 5). Right and left arrows are respectively in phase or in phase opposition, up and down arrows show respectively the first or the second signal which predates the other one.



**Figure 7** : (Top) Total solar irradiance variations (standardized over the whole period) from (Lean, 2009). (Bottom) Regression

5 (hPa/TSI std) of the sea-level pressure data from 20CR reanalysis over the TSI variations over the period 1870-2009 in spring (March, April, May, MAM). Only the values significant at the 95% level appear in colors, while the contours show all the computed values.