# Peer review of "Past African dust inputs in Western Mediterranean area controlled by the complex interaction between ITCZ, NAO and TSI"

_Climate of the Past, 2019_

## Referee Comment (RC1) · P. Sabatier et al. · 4 Oct 2019

The manuscript deals about a high-resolution record of Late Holocene Saharan dust supply into a mountainous lake on Corsica Island. The authors postulate the Intertropical Convergence Zone (ITCZ), the North Atlantic Oscillation (NAO) and solar irradiance (TSI) as major drivers for Saharan dust supply into the Western Mediterranean.

The manuscript provides generally an important issue as high-resolution records of Holocene Saharan dust emissions are rare and might be of interest for a broad international readership.

The presented study provides some excellent (pre-)conditions: the study site is well chosen. The lake is within a mountainous environment and provides a small catchment/lake surface ratio. These are very good pre-conditions for the reconstruction of a Saharan dust record. The catchment is free of carbonates. Therefore, hard-water effects do not influence 14C age control. The catchment consists of granodiorite. Therefore, catchment soil erosion is probably limited and the amount of Saharan dust of the silici-clastic fraction might be significant. Furthermore, the authors detect Palygorskite as clay mineral tracer in their record, generally indicating evidence for Saharan dust supply.

The study provides quantitative information about element contents in the core and in the lake's catchment, an important precondition for the use of element ratios as reliable proxies for Saharan dust input.

I have only few minor points:

Abstract

The abstract is generally well written.

Main text

Page 2 Line 5. What do you mean with Mt? Please explain the abbreviation, when you use it the first time.

Page 2 Line 16. please delete "In"

Page 3 Line 3. What do you mean with PSA? Please explain the abbreviation, when you use it the first time.

Page 3 line 15. Do you mean rubbles or screes? What is the genesis of the screes? Fluvial transport?

Page 5 line 22 dust proxy instead of Dust proxy

Page 6 line 17. Does the mean grain size (D50) correspond with remote aeolian dust? This should be discussed in the discussion chapter. You might discuss grain size endmembers as well.

Page 8 line 8. An enhanced I/K ratio...

Page 9 lines 13-18. This paragraph does not correspond to the subject of this chapter. You might delete this sentence.

Page 10 line 1. You may cite the new Lake Sidi Ali dust record here (2017, QSR) that show reduced dust supply into the Western Mediterranean during the AHP. The reference is already in your reference list.

Page 10 line 2. You mean increase in summer orbital insolation?

———————————————————

---

## Referee Comment (RC2) · Anonymous Referee #2 · 19 Nov 2019

Review of the paper "Past African dust inputs in Western Mediterranean area controlled by the complex interaction between ITCZ, NAO and TSI" by Sabatier et al. – Climate of the Past Discussion

General comments

The authors present a high-resolution mineralogical (XRD) and geochemical (XRF, ICPMS and ICP-OES) study of Lake Bastani sediments (Corsica). Sedimentological results help deciphering sediment sources and indicate that terrigenous supply in this high altitude lake is mainly composed of African dust supply over the last 3 millennia, and confirm that this Lake is an ideal place for reconstructing Saharan dust inputs

toward the west Mediterranean area over the last 3150 yrs cal. BP.

The well-constrained chronology (14C, 210Pb, 137Cs) of the core allows studying Saharan dust inputs with a millennial to centennial resolution using various statistical analyses (PCA, wavelet analysis, cross wavelet). The authors evidenced that long-term migration of the Intertropical Convergence Zone (ITCZ) appears to be the main forcing of the observed variations at millennial scale, probably affecting the dust availability over north and west Africa. During the last millennium, when the ITCZ is in its southward position, the North Atlantic Oscillation (NAO) seems to become the main climatic forcing, affecting both the pattern and the intensity of African dust transport. The authors also argued that the Total Solar Irradiance (TSI) was the main climatic forcing before 1070 yr cal BP when the ITCZ was in its northern position, through modifications of pressure gradient over Africa.

This paper deals with important scientific questions and highlights some new findings. The methods are adequate and assumptions are clearly outlined. The results mainly give support to the interpretations (see comments below) even if result tables are absent.

However some points are not completely clear, and minor corrections are needed:

1. The sediment sources paragraph (§4.1) is not totally convincing since some contrasting results are not discussed. Important data are missing, as the proportion of palygorskite in the 3 main potential source areas (PSA). Moreover, according to Figure 4:

- the I/K ratio mainly varies between 1 and 1.6, suggesting that Sahara is the main source of dust, pointing out PSA1 (1<I/K<2) as the main provenance (except ca. 2650 yr cal. BP, 2150 yr cal. BP, and between 900 and 700 yr cal. BP when I/K is below 1, suggesting some sahelian supply from PSA3 (0.3<I/K<0.7), and except ca. 2400 yr cal. BP and in the most recent part of the core when some influence of PSA2 (I/K>1.6) cannot be rule out);

- the C/K ratio varies between 0.2 and 0.8, suggesting western Sahara as the main source, with PSA2 (0<C/K<0.8) (and PSA3 with 0.2<C/K<0.9) being the main dust supplier. The C/K ratio reached 1 in the uppermost part of the core, suggesting potential contribution of PSA1 (C/K=1.5) - why PSA3 is ruled out as a contributor?

- the increase of palygorskite throughout the whole time interval may indicate enhanced contribution of sahelian source as PSA3 which is not consistent with the I/K and C/K ratio;

- How do you reconcile these apparently contrasting results (I/K indicating PSA1 as the main provenance, C/K suggesting PSA2 as the main source)?

- By the way, the main figure with sedimentological results (mineralogy and grain-size) is a supplementary material (Figure S1), which seems weird since the results should appear within the manuscript;

- Also the authors may consider adding the limit between Sahel and Sahara on figure 1;

2. Some of the geochemical data also evidenced contrasting results that deserve to be discussed thoroughly. I do agree that the Fe/Ca vs. Ti/Ca ratio indicate a good relationship between lake sediment and non-carbonated dust (figure 2b top), but this is not obvious when looking at the Fe/K vs. Ti/K ratio diagram: the lake samples plot on a line with a nice negative correlation while dust samples display a positive correlation! Please clarify.

3. The relationship between ITCZ and dust transport/emission is not new, but this study gives additional support using a statistical approach. Similarly the relationship between NAO and dust transport over the Mediterranean is not new but this study nicely documents the impact of NAO on dust input, and evidences NAO as a main forcing since 1070 yr cal. BP. The relationship between TSI and African dust input when the ITCZ is in its northern position is rather new, but as it is written, is not

completely convincing. I suggest the authors clarifying this paragraph and discussing the negative feedback of dust particles on irradiance.

Please also note the supplement to this comment:
https://www.clim-past-discuss.net/cp-2019-111/cp-2019-111-RC2-supplement.pdf
* * *
[Figure]

**Supplement:**

**Comments:**

Page 1, l. 27: The term "long-term" may be a bit confusing. I would recommend the authors to clearly specify in the manuscript what they mean by using "long-term"

Page 1, l. 31: explain the acronym for NAO

Page 2, l. 19-20: "the influence of the Holocene variability … remains poorly quantified": the authors may add some references as

> Cockerton et al., 2014, JGR – DOI:10.1002/2013JD021283
>
> Di Rita et al., 2018, Nature, Scientific Reports - DOI:10.1038/s41598-018-27056-2
>
> Hayes & Wallace, 2019, QSR - https://doi.org/10.1016/j.quascirev.2018.11.018

Page 3, l. 9: explain the acronym PSA

Page 3, l. 6: replace the coma after (Moulin et al., 1998) by a dot

Page 3, l. 1-2: I recommend the authors to mention in the abstract that the study is based on analytical approaches (including grain-size, mineralogy and geochemistry) as well as on statistical analysis

Page 3, l. 28: why is there a difference between the XRD sampling resolution (2 cm) compared with grain-size and LOI (1 cm)?

Page 5, l. 18: explain AR (Auto Regression?)

Page 6, l. 12: in the sentence " a high organic content", do you consider the LOI as reflecting the organic content? Does it mean that the uppermost part of the core contains up to 20% of organic carbon? The respective use of LOI550 and LOI950 is not very clear as described in the method page 3, l. 25-27.

I do not understand why the authors use both the LOI550 and NCIR on figure S1 since NCIR= initial dry weight - LOI550. Need clarification.

Sedimentological parameters of the uppermost part of the core seem rather different from the rest of the core: dark coloured (figure S1, left); slightly coarser grain-size; large range of variations for LOI550 and NCIR. Do you have any explanations?

Page 6, l. 16: You mentioned terrigenous supply from the watershed. Could they give some details about this supply?

Page 6, l. 16-18: "The grain-size presents a homogenous content (median (D50) = 32±8 μm". The average median is 32μm±8. Is it consistent with remote eolian supply? Add some references. On figure S1, the grain-size distribution is slightly different on the upper 10 cm. (see comment above)

Page 6, l. 19-20: I agree that chlorite and illite display an increasing trend throughout the record, but kaolinite seems to first increase between 100 and 50 meters, before varying around its average value between 50 and 20 meters. How do you explain this discrepancy between clay minerals?

Page 6, l. 20-21: How do you explain the peak in palygorskite around 6 cm, in phase with the NCIR% ?

Page 6, l. 24: I am not sure that I did understand the use of the composite section. The BAS13PA core seems to be the most detailed sequence and most analyses were performed on P4, but all radiocarbon analyses except one were performed on the BAS13P1. Please clarify.

Page 7, l. 8: (and Figure 2a) CaO seems to correlate with granodiorite watershed while $K_2O$ seems to correlate with Quaternary deposits. What is the main composition of these Quaternary deposits?

Page 7, l. 30-31: I would rather say, "This lake system is ideal for recording centennial variations of atmospheric inputs" according to the well-constrained chronology of the core

Page 8, l. 8-10: " The I/K ratio is typical of Saharan sources (so PSA1 or PSA2)..." "the low value of C/K is typical of western Sahara (PSA2)…"

But in details, according to figure 4:

- **The I/K ratio mainly varies between 1 and 1.6, suggesting that Sahara is the main source of dust, pointing out PSA1 (1<I/K<2) as the main provenance** (except ca. 2650 yr. cal. BP, 2150 yr. cal. BP, and between 900 and 700 yr. cal. BP when I/K is below 1, suggesting some sahelian supply from PSA3 (0.3<I/K<0.7), and except ca. 2400 yr. cal. BP and in the most recent part of the core when some influence of PSA2 (I/K>1.6) cannot be ruled out);

- **The C/K ratio varies between 0.2 and 0.8, suggesting western Sahara as the main source, with PSA2 (0<C/K<0.8) (and PSA3 with 0.2<C/K<0.9) being the main dust supplier.** The C/K ratio reached 1 in the uppermost part of the core, suggesting potential contribution of PSA1 (C/K=1.5) - why PSA3 is ruled out as a contributor?
- **The increase of palygorskite throughout the whole time interval may indicate enhanced contribution of sahelian source as PSA3** which is not consistent with the I/K and C/K ratio

How do you reconcile these apparently contrasting results (I/K indicating PSA1 as the main provenance, C/K suggesting PSA2 as the main source)?

Could you add the Sahel-Sahara limit on Figure 1?

Page 8, l. 10-13 "The decreasing trend in palygorskite (check the spelling a it appears as palygorskyte) content probably reflect progressive trend with more input from PSA1 over (? – A word is missing line 11), as also …". Why not considering PSA3 as a potential increasing palygorskite-depleted source? May be PSA1 is less rich in palygorskite compared with PSA2? Please add average percentages of palygorskite on figure 1

Page 8, l. 22-25: This sentence is a bit confusing. I suggest to modify "we used …" by "According to PCA analysis of the geochemical dataset, the ratio Fe/Ca vs. Ti/Ca and Fe/K vs. Ti/K were used in order to compare the compositions of lake sediments and associated watershed with dust deposits in NW Med…"

Page 8, l. 25-26: "These data allow to identify that geochemical composition of lake sediment is similar to non-carbonated dust samples" I do agree that the Fe/Ca vs. Ti/Ca ratio indicate a good relationship between lake sediment and non-carbonated dust (figure 2b top), but this is not obvious when looking at the Fe/K vs. Ti/K ratio diagram: the lake samples plot on a line with a nice negative correlation while dust samples display a positive correlation! Please clarify the sentence line 25-26

Page 8, l. 28: "Moreover, Fe and Ca, K contents present the same variations in lake sediments" – data not shown? I agree for Ca, but I am not convinced for K

Page 9, L. 10: "significant increase from approximately 1000 yr. cal. BP" There is one peak in the Fe signal around 1000 yr. cal. BP but if ignoring this peak, it seems that the enhanced supply in Fe started around 700 yr. cal. BP.

Page 9, l.14: "we observed an African dust increase that reached …" In the Fe record? In previously published data (give refs)?

Page 9, l. 21: I suggest adding Northern Hemisphere insolation record on figure 4 next to the ITCZ variations

Page 9, l. 24: I agree with the idea of the relationship between the southward position of the ITCZ and dust emission event if Doherty et al., 2012, 2014 mainly deal with dust transport rather than with dust emission

Page 10, l. 12: "ENSO" not shown?

Page 10, l. 13-14: the period seems to be slightly higher for dust compare with ITCZ

Figure 5: why the y-scale is different for NAO+ (seems to be vertically compressed)? Could you align the x-scale of the diagram in order to make the comparison of NAO+ with the other data?

Page 10, L. 20 and Figure 6: what is the period for cross wavelet analyses for dust input vs. NAO? It seems to be around 450-500 yr.? What is the robustness of this period considering the length of the analysed record (1000 yr.)?

Page 10, l. 30-31: "which correspond to the **period** of long-term increase in African dust " Do you mean time-interval, "suggesting that the long-term forcing through ITCZ migration have an impact on the NAO/African dust correlation" I am not convinced that your dataset evidence that the progressive southward migration of the ITCZ has an impact on the NAO/dust correlation. Could you clarify?

Page 10, l. 34: "The position of Westerlies are influenced by the NAO" indeed, modification of the Westerlies is one of the consequences of the north Atlantic Oscillation (Moulin et al., 1997)

Page 11, l. 2: The positive phase of the NAO is modelled in winter but with an impact on ITCZ during spring (april)

Page 11, l. 14: "large changes in solar radiation" What is the range of variation over the last 3000 yr.? Does this range of variations large enough to promote the proposed modifications of the climatic system? Could you consider adding a record of calculated irradiance over the last 3000 yr.?

Page 11, l. 25: "strong negative anomaly" I do not get it! Could you give additional details on order to highlight this negative anomaly on figure 7?

Page 11, l. 17-34: I am not convinced by this paragraph since I am not able to evaluate the potential impact of changes in irradiance on dust emission. May be of interest to have a look on studies about the impact of dust on surface solar irradiance since the emission of dust may modify the effect of solar irradiance on surface

Kosmopoulos et al., 2017, Atmos. Meas. Tech., https://doi.org/10.5194/amt-10-2435-2017

Alonso-Montesinos et al., 2017, Atmospheric Environment, https://doi.org/10.1016/j.atmosenv.2017.09.040

Granados-Munoz et al., 2019, Atmos. Chem. Phys., https://doi.org/10.5194/acp-19-523-2019

Page 12, l. 8: I am not convinced by the conclusion "with an increase since 1070 yr. cal. BP in response to a gradual orbitally-induced decrease in northern Hemisphere insolation". I do agree about the long-term relationship (over the last 3000 yr.) but I disagree with the 1070 yr. inflexion point

Page 12, l. 11: "since 1070 yr. cal. BP, the NAO is dominant…". The wavelet analysis was performed over the last 2000 yr., so you cannot rule out any influence of the NAO on the 3000-2000 yr. interval

**Typo:**

Page 2, l. 16: delete the extra "in" before "At the scale…"

Page 3, l. 6: replace the coma after (Moulin et al., 1998) by a dot

Page 4, l. 28: delete the extra bracket after « Aitchison et al., 2002)) »

Page 4, l. 30: delete the extra brackets "((van den Boogaart and Tolosana-Delgado, 2008))"

Page 5, l. 22: replace "Dust" by "dust"

Page 6, l. 25: delete the extra ";" after the bracket

Page 8, l. 21: I suggest to replace "with" by ":"; "while" by "and" and "watershed is" by "watershed samples are"

Page 8, l. 24: add a coma in between Sicily and Tomadin (also in figure 2, Page 21 l. 8)

Page 13, l. 10: delete the extra « ' » after « Barcelo' »

Page 13, l. 13: add a space before « Berlin »

Page 15, l. 23: use lowercase for the title

Page 14, l. 26: use lowercase for the title

Page 16, l. 16: change "M.D. Loÿe-Pilot" by "Loÿe-Pilot, M.D.", also Page 8, l. 15-16

Choose between « millennia » Page 1, l. 17 and Page 10, l. 17 and « millenniums » Page 10, l. 8

Check the spelling of palygorskite throughout the manuscript since it appears sometimes as palygorskyte

**Figures:**

Figure 1: Can you consider adding major winds and palygorskite percentages? Indeed the palygorskite varies between circa 5 to 20% and the PSA mineralogical signatures would help to interpret the observed variations. If palygorskite is <5% in PSA3, then you need to consider some contribution from this source area in the uppermost part of the core

Figure 2: symbols (brown diamonds and orange squares) on figure 2b are hardly readable; can you consider having the figure 2c in an individual supplementary figure?

Figure 3: the blue symbols are not readable

Figure 4: replace "palygorskyte" by "palygorskite"; add mean value for palygorskite

Figure 5: use same x- and y-scale for all 4 parameters (NAO+); check the legend Page2, l. 3-4: "labelled in white" since I do not see white labels

Figure S1: replace "palygorskyte" by "palygorskite"

---

## Short Comment (SC1) · 27 Nov 2019

In Fig. 4, why did you not use the d18O record of monsoon rainfall and ITCZ position over West Africa from Shanahan et al. (2009) to discuss the influence of the ITCZ migration on your record in the Western Mediterranean ?

Shanahan, T.M., Overpeck, J.T., Anchukaitis, K.J., Beck, J.W., Cole, J.E., Dettman, D.L., Peck, J.A., Scholz, C.A., King, J.W., 2009. Atlantic forcing of persistent drought in West Africa. Science 324, 377-380.

---

## Author Comment (AC1) · 27 Nov 2019

Dear Jean-Philippe Degai,

For ITCZ proxy we prefer use the Ti curve from Haug et al., (2001) because this one is more integrative than the one of Shanahan et al., (2009). ITCZ reconstitution from Cariaco basin (Haug et al., 2001) in related to Orinoco river input a â $\breve{A}$ Ő880,000 km2 wide catchment, while the African monsoon reconstitution from Shanahan et al., (2009) was from a crater lake (Lake Bosumtwi). Thus, for this last reconstitution local effect could have a greater impact. Moreover, the reconstitution for Lake Bosumtwi cover just the last 2600 yrs while we present here a 3150 yrs archive of African dust record.

[Figure]

However, this two ITCZ reconstitutions present many similarities in term of long and short term variations, even at decadal scale (Shanahan et al., 2009).

Haug, G. H., Hughen, K., Sigman, D. M., Peterson, L. C. and Röhl, U.: Southward Migration of the Intertropical Convergence Zone Through the Holocene, Science, 293(5533), 1304–1308, doi:10.1126/science.1059725, 2001.

Shanahan, T.M., Overpeck, J.T., Anchukaitis, K.J., Beck, J.W., Cole, J.E., Dettman,D.L., Peck, J.A., Scholz, C.A., King, J.W., 2009. Atlantic forcing of persistent drought inWest Africa. Science 324, 377-380.

---

## Author Comment (AC2) · 27 Nov 2019

We acknowledge the anonymous referee#1 for this kind review. We revised it in detail as described below.

Page 2 Line 5. What do you mean with Mt? Please explain the abbreviation, when you use it the first time. Metric Tonne

Page 2 Line 16. please delete "In" Thank

Page 3 Line 3. What do you mean with PSA? Please explain the abbreviation, when you use it the first time. Potential Source Area

[Figure]

Page 3 line 15. Do you mean rubbles or screes? What is the genesis of the screes? Fluvial transport? Screes from glacial or rock fall origins

Page 5 line 22 dust proxy instead of Dust proxy Thanks

Page 6 line 17. Does the mean grain size (D50) correspond with remote aeolian dust? This should be discussed in the discussion chapter. You might discuss grain size end-members as well. Grain size in this lake sediment is mostly affected by biogenic silica (see below) thus this data could not be used to track thin particle from Aeolian origin. We just use here grain size to illustrate that there is not coarse deposit link to flood or terrigenous events

Page 8 line 8. An enhanced I/K ratio… Thanks

Page 9 lines 13-18. This paragraph does not correspond to the subject of this chapter. You might delete this sentence. Thank, we move this paragraph to the part "4.2.2." about Centennial variation

Page 10 line 1. You may cite the new Lake Sidi Ali dust record here (2017, QSR) that show reduced dust supply into the Western Mediterranean during the AHP. The reference is already in your reference list. We add this reference

Page 10 line 2. You mean increase in summer orbital insolation? Thank, Yes increase for the HAP

[Figure]

[Figure]

**Fig. 1.**

---

## Author Comment (AC3) · 9 Dec 2019

We acknowledge the anonymous referee#2 for this constructive review. We revised it in detail as described below from the document in supplementary which summurize major and minor coments.

detailed Comments: Page 1, l. 27: The term "long-term" may be a bit confusing. I would recommend the authors to clearly specify in the manuscript what they mean by using "long-term" In the same phrase me precise Âń Millennial Âż

Page 1, l. 31: explain the acronym for NAO We add

Page 2, l. 19-20: "the influence of the Holocene variability ... remains poorly quantified": the authors may add some references as Cockerton et al., 2014, JGR – DOI:10.1002/2013JD021283 > We add this reference thank Di Rita et al., 2018, Nature, Scientific Reports - DOI:10.1038/s41598-018-27056-2 > We do not see any link with this publication Hayes & Wallace, 2019, QSR - https://doi.org/10.1016/j.quascirev.2018.11.018 > This publication is a synthesis and use previously publish dust reconstitution, thus we prefer cited the related reference for Holocene variability > (Mulitza et al., 2010)

Page 3, l. 9: explain the acronym PSA Thank we add : potential source area (PSA)

Page 3, l. 6: replace the coma after (Moulin et al., 1998) by a dot OK

Page 3, l. 1-2: I recommend the authors to mention in the abstract that the study is based on analytical approaches (including grain-size, mineralogy and geochemistry) as well as on statistical analysis Mineralogical, geochemical and wavelet analyses are already mention in the abstract (P1, L21, 30)

Page 3, l. 28: why is there a difference between the XRD sampling resolution (2 cm) compared with grain-size and LOI (1 cm)? Because this analytical method is time consuming with chemical long chemical preparation, thus we choose to analyse less samples compare to faster acquired analysis such as grain-size and LOI

Page 5, l. 18: explain AR (Auto Regression?) Yes Auto Regressive, we add: Auto Regressive model (first order: AR(1))

Page 6, l. 12: in the sentence " a high organic content", do you consider the LOI as reflecting the organic content? Does it mean that the uppermost part of the core contains up to 20% of organic carbon? The respective use of LOI550 and LOI950 is not very clear as described in the method page 3, l. 25-27. I do not understand why the authors use both the LOI550 and NCIR on figure S1 since NCIR= initial dry weight - LOI550. Need clarification. LOI550 and LOI950 represent respectively the organic and

carbonate content of a sample express in %. The NCIR is the non-carbonate ignition residue which correspond mostly correspond to silicate. We add this sentence in the method part LOI550 is not equal to Organic Carbon because during Organic burning CO and CO2 are created and we do not a monitoring of these molecules ratio. We use these two curves for a better reading of this figure even on is the opposite of the other

Sedimentological parameters of the uppermost part of the core seem rather different from the rest of the core: dark coloured (figure S1, left); slightly coarser grain-size; large range of variations for LOI550 and NCIR. Do you have any explanations? Yes, this explanation is previously publish, we add the reference to this work by adding the following sentence: Âń This upper increase in organic matter content is related to the lake eutrophication induced by recent increase of atmospheric nitrogen deposit which enhance lake primary productivity (Roche and Loÿe-Pilot, 1989)."

Page 6, l. 16: You mentioned terrigenous supply from the watershed. Could they give some details about this supply? In this result part we just hypothesis that the NCIR is probably due to biogenic silica and terrigenous input from the watershed and eolian deposit and thus in the discussion we conclude than the input from the watershed is negligible. Thus we do not have any detail to provide on this supply because we do not observe any watershed input.

Page 6, l. 16-18: "The grain-size presents a homogenous content (median (D50) = $32\pm8$ $\mu$m". The average median is $32\mu$m$\pm8$. Is it consistent with remote eolian supply? Add some references. On figure S1, the grain-size distribution is slightly different on the upper 10 cm. (see comment above) Grain size in this lake sediment is mostly affected by biogenic silica (see figure in the answer of Reviewer 1 comment) thus these grain size data could not be used to track thin particle from Aeolian origin. We just use here grain size to illustrate that there is not coarse deposit link to flood or terrigenous events. For the upper centimetre it is probably due to lake eutrophisation with larger Diatom, see above comment. We add the following sentence: Âń Grain size. . . is mostly affected by biogenic silica not remove before analysis. However, no

large grain size variation related to flood events (Sabatier et al., 2017; Wilhelm et al., 2015) Âż

Page 6, l. 19-20: I agree that chlorite and illite display an increasing trend throughout the record, but kaolinite seems to first increase between 100 and 50 meters, before varying around its average value between 50 and 20 meters. How do you explain this discrepancy between clay minerals? In the text me add "... with a stabilization for kaolinite around 50cm Âż, In this result part we do not want to interpret the clay mineral profile, the interpretation is provide in the discussion part

Page 6, l. 20-21: How do you explain the peak in palygorskite around 6 cm, in phase with the NCIR% ? Probably a short term input of African dust, this peak is also correlated to the Fe proxy of dust, but again in this result part we do not what to interpret the result.

Page 6, l. 24: I am not sure that I did understand the use of the composite section. The BAS13PA core seems to be the most detailed sequence and most analyses were performed on P4, but all radiocarbon analyses except one were performed on the BAS13P1. Please clarify. We used a composite section because the longer core P4 is sample with hammering thus we can observe a more compacted part over the upper section related to high water content, thus we prefer use for the upper part a better preserved section. The P3 is too short for long term reconstitution. For C14 age it is quite difficult to find macroremain thus we different core to find enough material and then we correlate age on the composite core thanks to XRF data

Page 7, l. 8: (and Figure 2a) CaO seems to correlate with granodiorite watershed while K2O seems to correlate with Quaternary deposits. What is the main composition of these Quaternary deposits? Unfortunately, we do not have a full description of these deposits, but there are moraine and delta deposits with some organic material and me by more thin material. But it will be too much speculative to interpret is information. The important thing is that lake sediment is low influenced by CaO and K2O as illustrated

in figure 2 and discuss in the manuscript

Page 7, l. 30-31: I would rather say, "This lake system is ideal for recording centennial variations of atmospheric inputs" according to the well-constrained chronology of the core I'm not fully agree with this, because if this lake sediment record well define lead pollution coming from atmospheric deposition it mean that this lake is also good to track atmospheric input and yes the chronology is also well constrain

Page 8, l. 8-10: " The I/K ratio is typical of Saharan sources (so PSA1 or PSA2)..." "the low value of C/K is typical of western Sahara (PSA2)..." But in details, according to figure 4: - The I/K ratio mainly varies between 1 and 1.6, suggesting that Sahara is the main source of dust, pointing out PSA1 (1<I/K<2) as the main provenance (except ca. 2650 yr. cal. BP, 2150 yr. cal. BP, and between 900 and 700 yr. cal. BP when I/K is below 1, suggesting some sahelian supply from PSA3 (0.3<I/K<0.7), and except ca. 2400 yr. cal. BP and in the most recent part of the core when some influence of PSA2 (I/K>1.6) cannot be ruled out); - The C/K ratio varies between 0.2 and 0.8, suggesting western Sahara as the main source, with PSA2 (0<C/K<0.8) (and PSA3 with 0.2<C/K<0.9) being the main dust supplier. The C/K ratio reached 1 in the uppermost part of the core, suggesting potential contribution of PSA1 (C/K=1.5) - why PSA3 is ruled out as a contributor? Thanks for this constructive comment, Yes, the I/K ratio of the record (around 1.2) is typical from PSA1 (11.6) and PSA3 (0.3<I/K<0.7) if there are multiple sources contribution. C/K ratio is typical of PSA2 (0<C/K<0.8) or PSA3 (0.2<C/K<0.9) and far from PSA1 (C/K=1.5), except for the upper part of the record. Thus if we consider these two ratios together the main dust sources are PSA2 and PSA3 except for the recent part. So we thank the reviewer for this comment and modify this part according to this interpretation in the new version of the manuscript.

- The increase of palygorskite throughout the whole time interval may indicate enhanced contribution of sahelian source as PSA3 which is not consistent with the I/K and C/K ratio How do you reconcile these apparently contrasting results (I/K indicating

PSA1 as the main provenance, C/K suggesting PSA2 as the main source)? Could you add the Sahel-Sahara limit on Figure 1? We add the Sahara/Sahel limit on figure 1 The review probably wants to said "the decrease of palygorskite" as the long term trend over the 3000 ys record present a decreasing trend. Palygorskite content in variable between each potential sources, PSA3 is clearly depleted while PSA1 and PSA2 are rich in palygorskite (Formenti et al., 2011; Scheuvens et al., 2013). However, there are large variations in palygorskite content in PSA1 and PSA2, as these PSA are large areas. For Grousset et al., (1992) and Bout-Roumazeilles et al., (2013), PSA2 (W Sahara) seem more rich in palygorskite than PSA1 (N Sahara). Thus, we interpret this decreasing trend in palygorskite as a result of a more important influence from PSA1 through time, attested by the increasing trend of C/K ratio and higher value for the upper part of the record. In the new version of the manuscript we add the reference to the work of Grousset et al., (1992) to support this long term interpretation.

Page 8, l. 10-13 "The decreasing trend in palygorskite (check the spelling a it appears as palygorskyte) content probably reflect progressive trend with more input from PSA1 over (? – A word is missing line 11), as also . . .". Why not considering PSA3 as a potential increasing palygorskite-depleted source? May be PSA1 is less rich in palygorskite compared with PSA2? Please add average percentages of palygorskite on figure 1 Thanks we verify the spelling of palygorskite over the whole manuscript. Yes a part of the sentence is missing, we add "over the last 3150 yrs cal BP" and it is in agreement with higher PSA1 supply in regard to higher C/K and I/K ratio. See above comment for palygorskite interpretation. In figure 1, we prefer keep the current notation for palygorskite contents (Paly+ for PSA1 and PSA2, Paly- for PSA3) because precise palygorskite percentages are variables.

Page 8, l. 22-25: This sentence is a bit confusing. I suggest to modify "we used . . ." by "According to PCA analysis of the geochemical dataset, the ratio Fe/Ca vs. Ti/Ca and Fe/K vs. Ti/K were used in order to compare the compositions of lake sediments and associated watershed with dust deposits in NW Med. . ." Thanks

Page 8, l. 25-26: "These data allow to identify that geochemical composition of lake sediment is similar to noncarbonated dust samples" I do agree that the Fe/Ca vs. Ti/Ca ratio indicate a good relationship between lake sediment and non-carbonated dust (figure 2b top), but this is not obvious when looking at the Fe/K vs. Ti/K ratio diagram: the lake samples plot on a line with a nice negative correlation while dust samples display a positive correlation! Please clarify the sentence line 25-26 We used this diagram to illustrate that geochemical signature of lake sediment is in agreement with dust signature. To facilitate the reading of this figure we add an orange area corresponding to African dust signature from available bibliography data and lake sediment samples are present in this area in both panel. We do not try to make any proportion calculation or correlation; we just compare geochemical signature in term of possible signature. The negative correlation for lake sediment is not so obvious including samples uncertainties. We add a sentence to clarify this

Page 8, l. 28: "Moreover, Fe and Ca, K contents present the same variations in lake sediments" – data not shown? I agree for Ca, but I am not convinced for K We add the correlation coefficient: Fe vs Ca r=0.63, p < 10-16; Fe vs K r=0.63, p < 10-16 It is also visible on the PCA in Figure S3

Page 9, L. 10: "significant increase from approximately 1000 yr. cal. BP" There is one peak in the Fe signal around 1000 yr. cal. BP but if ignoring this peak, it seems that the enhanced supply in Fe started around 700 yr. cal. BP. No, with the eyes the baseline without peak increases from around 1000 cal BP and the breackpoint statistical analysis identify also arourn 1000 cal BP

Page 9, l.14: "we observed an African dust increase that reached . . ." In the Fe record? In previously published data (give refs)? Yes in Fe data in agreement with the publication of Evan and Mukhopadhyay, (2010). To precise this statement me add Âń in Fe data"

Page 9, l. 21: I suggest adding Northern Hemisphere insolation record on figure 4 next

to the ITCZ variations In the Figure 4 we add the 20°N June insolation

Page 9, l. 24: I agree with the idea of the relationship between the southward position of the ITCZ and dust emission event if Doherty et al., 2012, 2014 mainly deal with dust transport rather than with dust emission Thanks

Page 10, l. 12: "ENSO" not shown? We precise that the wavelet analyses is not shown in this paper Page 10, l. 13-14: the period seems to be slightly higher for dust compare with ITCZ Figure 5: why the y-scale is different for NAO+ (seems to be vertically compressed)? Could you align the xscale of the diagram in order to make the comparison of NAO+ with the other data? Yes we change this figure and we align X and Y on the same scale. The period around 2700-3100 years is not well constrain in age for ITCZ (120-250) but match with the one of dust signal

Page 10, L. 20 and Figure 6: what is the period for cross wavelet analyses for dust input vs. NAO? It seems to be around 450-500 yr.? What is the robustness of this period considering the length of the analysed record (1000 yr.)? We thank the reviewer for this point. We forgot to precise that because of wavelet analysis, Yaxis is dyadic. More accurately, the main period is around 415 years but ranges from 350 to 470. This is longer than the periods published in literature for NAO. The meaning of wavelet coherency is not directly linked to wavelet structure of individual series but to the covariation between each scale. A speculative interpretation should be that there is a buffering between the trigger (NAO) and the effect (Dust transport) explaining the discrepancy between NAO known periods and this wavelet coherency. About the robustness of this period, this analysis is only interpreted in the cone of interest (COI) where no edge effect could interfere the wavelet analysis then the coherency analysis. Indeed, such analyses couldn't have been done with FFT analysis because of FFT based assumptions

Page 10, l. 30-31: "which correspond to the period of long-term increase in African dust " Do you mean timeinterval, "suggesting that the long-term forcing through ITCZ

migration have an impact on the NAO/African dust correlation" I am not convinced that your dataset evidence that the progressive southward migration of the ITCZ has an impact on the NAO/dust correlation. Could you clarify?

Page 10, l. 34: "The position of Westerlies are influenced by the NAO" indeed, modification of the Westerlies is one of the consequences of the north Atlantic Oscillation (Moulin et al., 1997) Yes for sure, but for the demonstration it is important to call back this.

Page 11, l. 2: The positive phase of the NAO is modelled in winter but with an impact on ITCZ during spring (april) The NAO reconstruction is based on different proxies that are not specifically sensitive to winter. Thus, the Franke et al. (2017) NAO reconstruction is annual, including also atmospheric variation during the other seasons that resemble NAO-like structure. This is a classical approximation in NAO reconstructions. For present-day, annual NAO are mainly dominated by winter variations, but the other seasons can also have an influence. Thus, we argue that this may explain the linkage we have found between annual NAO and spring ITCZ.

Page 11, l. 14: "large changes in solar radiation" What is the range of variation over the last 3000 yr.? Does this range of variations large enough to promote the proposed modifications of the climatic system? Could you consider adding a record of calculated irradiance over the last 3000 yr.? The range of variation of the solar radiation are subject to strong debate. The main consensus is that the amplitude of centennial variability may be of the same order of magnitude as the well-observed 11-year cycle. In that sense, the regression analysis we produced in Fig. 7 is well-adapted to show that such an amplitude is likely sufficient to have a significant impact on the climatic system, even if very weak. Thus, we agree with the reviewer that the changes are not that large, and we have removed this adjective in the manuscript. Indeed, the former sentences were rather insisting on the UV part of the spectrum which is more changing than the whole TSI, so that the changes in TSI are usually moderate, while changes in UV can be larger. We do not feel that adding a record of calculated irradiance will

bring much, since we are mainly looking at qualitative relationship in this part of the manuscript, and are not using climate models there.

Page 11, l. 25: "strong negative anomaly" I do not get it! Could you give additional details on order to highlight this negative anomaly on figure 7? Figure 7 of the initial manuscript is showing the regression of SLP on TSI. Thus, we have positive values when TSI is increasing, which are shown in the Figure. When TSI is decreasing (what is described in the sentence), the anomalies are getting negative, as discussed here. Nevertheless, we agree that this may be misleading since we are mainly discussing negative TSI anomalies in the paper, so we have reversed the sign of the regression to make things easier to follow and we have modified the figure as follow:

Page 11, l. 17-34: I am not convinced by this paragraph since I am not able to evaluate the potential impact of changes in irradiance on dust emission. May be of interest to have a look on studies about the impact of dust on surface solar irradiance since the emission of dust may modify the effect of solar irradiance on surface Kosmopoulos et al., 2017, Atmos. Meas. Tech., https://doi.org/10.5194/amt-10-2435-2017 Alonso-Montesinos et al., 2017, Atmospheric Environment, https://doi.org/10.1016/j.atmosenv.2017.09.040 Granados-Munoz et al., 2019, Atmos. Chem. Phys., https://doi.org/10.5194/acp-19-523-2019 We agree with the reviewer that the mechanism we are proposing is not supported by simulations including the dust emission and transport (although it is supported by a modelling paper concerning the link between TSI and atmospheric circulation). We have added a discussion in the manuscript to highlight this caveat and we also discussed the potential feedback of dust emission on solar irradiance reaching the surface, using the three papers mentioned by the reviewer.

Page 12, l. 8: I am not convinced by the conclusion "with an increase since 1070 yr. cal. BP in response to a gradual orbitally-induced decrease in northern Hemisphere insolation". I do agree about the long-term relationship (over the last 3000 yr.) but I disagree with the 1070 yr. inflexion point The break-point analyses on the linear

regression model between reconstructed dust signal and ITCZ clearly identify that 1070 yr is the main change with a broken-line relationships. To integrate the reviewer caution we change "have been forced" by "could be forced" in the new version of the manuscript

Page 12, l. 11: "since 1070 yr. cal. BP, the NAO is dominant...". The wavelet analysis was performed over the last 2000 yr., so you cannot rule out any influence of the NAO on the 3000-2000 yr. interval Ok, we moderate this sentence: Even if precise NAO reconstitution is not available before 2000, wavelet analysis show that between 3150 and 1070 yr cal BP (ITCZ in more northern position) the centennial increases of Saharan dust inputs are correlated to low TSI.

Typo: Page 2, l. 16: delete the extra "in" before "At the scale..."

Page 3, l. 6: replace the coma after (Moulin et al., 1998) by a dot

Page 4, l. 28: delete the extra bracket after Âń Aitchison et al., 2002)) Âż

Page 4, l. 30: delete the extra brackets "((van den Boogaart and Tolosana-Delgado, 2008))"

Page 5, l. 22: replace "Dust" by "dust"

Page 6, l. 25: delete the extra ";" after the bracket

Page 8, l. 21: I suggest to replace "with" by ":"; "while" by "and" and "watershed is" by "watershed samples are"

Page 8, l. 24: add a coma in between Sicily and Tomadin (also in figure 2, Page 21 l. 8)

Page 13, l. 10: delete the extra Âń ' Âż after Âń Barcelo' Âż

Page 13, l. 13: add a space before Âń Berlin Âż

Page 15, l. 23: use lowercase for the title

Page 14, l. 26: use lowercase for the title

Page 16, l. 16: change "M.D. Loÿe-Pilot" by "Loÿe-Pilot, M.D.", also Page 8, l. 15-16

Choose between Âń millennia Âż Page 1, l. 17 and Page 10, l. 17 and Âń millenniums Âż

Page 10, l. 8 Check the spelling of palygorskite throughout the manuscript since it appears sometimes as palygorskyte Thank, we correct all these typo

Figures: Figure 1: Can you consider adding major winds and palygorskite percentages? Indeed the palygorskite varies between circa 5 to 20% and the PSA mineralogical signatures would help to interpret the observed variations. If palygorskite is <5% in PSA3, then you need to consider some contribution from this source area in the uppermost part of the core In figure 1, we prefer keep the current notation for palygorskite contents (Paly+ for PSA1 and PSA2, Paly- for PSA3) because precise palygorskite percentages are variables (Bout-Roumazeilles et al., 2013; Grousset et al., 1992). For sources contributions and palygorskite variation over the upper part see above comment.

Figure 2: symbols (brown diamonds and orange squares) on figure 2b are hardly readable; can you consider having the figure 2c in an individual supplementary figure? We modify the symbols colours: Orange > light orange, brown > dark brown

Figure 3: the blue symbols are not readable We enlarge the blue symbol

Figure 4: replace "palygorskyte" by "palygorskite"; add mean value for palygorskite Thanks Figure 5: use same x- and y-scale for all 4 parameters (NAO+); check the legend Page2, l. 3-4: "labelled in white" since I do not see white labels We modify this figure to have the same X and Y scale and change the label white to black, thanks

Figure S1: replace "palygorskyte" by "palygorskite" Thanks

References: Bout-Roumazeilles, V., Combourieu-Nebout, N., Desprat, S., Siani, G., Turon, J.-L. and Essallami, L.: Tracking atmospheric and riverine terrigenous supplies variability during the last glacial and the Holocene in central Mediterranean, Climate of

the Past, 9(3), 1065–1087, doi:10.5194/cp-9-1065-2013, 2013.

Evan, A. T. and Mukhopadhyay, S.: African Dust over the Northern Tropical Atlantic: 1955–2008, Journal of Applied Meteorology and Climatology, 49(11), 2213–2229, doi:10.1175/2010JAMC2485.1, 2010.

Formenti, P., Schütz, L., Balkanski, Y., Desboeufs, K., Ebert, M., Kandler, K., Petzold, A., Scheuvens, D., Weinbruch, S. and Zhang, D.: Recent progress in understanding physical and chemical properties of African and Asian mineral dust, Atmospheric Chemistry and Physics, 11(16), 8231–8256, doi:10.5194/acp-11-8231-2011, 2011.

Grousset, F. E., Rognon, P., Coudé-Gaussen, G. and Pédemay, P.: Origins of peri-Saharan dust deposits traced by their Nd and Sr isotopic composition, Palaeo-geography, Palaeoclimatology, Palaeoecology, 93(3–4), 203–212, doi:10.1016/0031-0182(92)90097-O, 1992.

Mulitza, S., Heslop, D., Pittauerova, D., Fischer, H. W., Meyer, I., Stuut, J.-B., Zabel, M., Mollenhauer, G., Collins, J. A., Kuhnert, H. and Schulz, M.: Increase in African dust flux at the onset of commercial agriculture in the Sahel region, Nature, 466(7303), 226–228, doi:10.1038/nature09213, 2010.

Roche, B. and Loÿe-Pilot, M. D.: Eutrophisation récente d'un lac de montagne sans occupation humaine (lac de Bastani, Corset­Ăŕ: Conséquence d'agents atmo-sphériques?, Revue des sciences de l'eau, 2(4), 681–707, doi:10.7202/705049ar, 1989.

Sabatier, P., Wilhelm, B., Ficetola Gentile, F., Moiroux, F., Poulenard, J., Develle, A.-L., Bichet, A., Chen, W., Pignol, C., Reyss, J.-L., Gielly, L., Bajard, M., Perrette, Y., Malet, E., Taberlet, P. and Arnaud, F.: 6-kyr record of flood frequency and intensity in the western Mediterranean Alps – Interplay of solar and temperature forcing, Quaternary Science Reviews, 170, 121–135, doi:10.1016/j.quascirev.2017.06.019, 2017.

Scheuvens, D., Schütz, L., Kandler, K., Ebert, M. and Weinbruch, S.: Bulk composition

of northern African dust and its source sediments — A compilation, Earth-Science Reviews, 116, 170–194, doi:10.1016/j.earscirev.2012.08.005, 2013.

Wilhelm, B., Sabatier, P. and Arnaud, F.: Is a regional flood signal reproducible from lake sediments?, Sedimentology, 62(4), 1103–1117, doi:10.1111/sed.12180, 2015.

[Figure]

a) Solar variations

b) Inverse regression SLP (MAM) over solar variations

**Fig. 1.**

---

## Author Response (AR2)

RESPONSE TO EDITOR COMMENTS

Pierre Sabatier

Laboratoire Environnement Dynamiques et Territoire de Montagne

UMR 5204 CNRS - Université de Savoie
Bât. Pôle Montagne
73370 Le Bourget du Lac Cedex, FRANCE

pierre.sabatier@univ-savoie.fr

Dear Nathalie Combourieu-Nebout,

Thank you for your support for this manuscript publication in Climate of the Past, we have made these last changes in the revised manuscript comments is provided in red below and in the new version of the manuscript.

Best regards
Pierre Sabatier

1- Please follow the Rev. 1 and add something about screes I did not seen that.
We add in the revised version of the manuscript « from glacial or rock fall origins »

2- Concerning the remark on page 6 line 17 of Rev.1 on the grain size, why didn't you add your remark in the text, it will be informative. Please do that.

We add « Grain size in this lake sediment is mostly affected by biogenic silica (see below) thus this data could not be used to track thin particle from Aeolian origin. We just use here grain size to illustrate that there is not coarse deposit link to flood or terrigenous events »

3- I agree with the reviewer 2 and ask you to precise what you call long term as this term is used for long records to explain the large changes and in fact for lot of researchers millennial changes are called short-term changes. Please modify or explain more.

We add this sentence in the asbstract : « High resolution geochemical contents provide a reliable proxy of Saharan dust inputs with long term (millennial) to short term (centennial) variations »

At the end of the introduction we also add : « long term (millennial) to short term (centennial) variations »

4- I think that something has to be added concerning the remark of reviewer 2 on the composite section and the other section. Perhaps one or two sentences, may be in the "study area" §.
We probably not well formulate this part, now we add in the method part « These 3 cores were correlated and we used data from different sections to have enough material for different analyses. And we remove the term composite to avoid any confusion.

5- Concerning the palygorskite, I agree with reviewer 2 that values may be better in the figure 1 as it has been done for the ratios. You may add range of values if necessary. Please try to do that.

We add a range of palygorskite content in figure 1

6- In the same figure choose another colour to represent the Sahara/Sahel limit please. Orange is not very visible on the grey background.

We change orange to blod red in figure 1

7- In the Figure 7 You said thet you aligned the scales but the 0 is not aligned here especially for the TSI and we do not see the marks that indicate the different ages. Please add them and change that. You probably mean for Figure 5, thanks the number 0 is not well aligned we correct this, but for the graph scales it is ok.

8- In response to the remark of Rev.2 on page 11 l.17-34 of your first version, I see nothing in the new text although you said that you added sentences. Perhaps it has been added elsewhere.

We first do not think that we need to add a sentence to support our demonstration, but according to your recommendation we add in the paragraph two new sentences related to our R2 response:

« To support this link between the TSI and changes in pressure pattern over North Africa, as the amplitude of centennial variability may be of the same order of magnitude as the well-observed 11-year cycle…. »

and

« Even if TSI variations are weak, this regression analysis show that such an amplitude is likely sufficient to have a significant impact on the climatic system »

9- A little remark on the Mt symbol in the introduction. Perhaps an explanation is necessary in the text for the reader as requested by Rev. 2.

We add « Metric Tonne »